# Risk of 28-day readmissions among stroke patients in Malaysia (2008–2015): Trends, causes and its associated factors

**Swee Hung Ang**[1¤]*, **Wen Yea Hwong**[1,2], **Michiel L. Bots**[2], **Sheamini Sivasampu**[1], **Aznida Firzah Abdul Aziz**[3], **Fan Kee Hoo**[4], **Ilonca Vaartjes**[2]

**1** Institute for Clinical Research, National Institutes of Health, Ministry of Health, Selangor, Malaysia, **2** Julius Center for Health Sciences and Primary Care, University Medical Center Utrecht, Utrecht University, Utrecht, The Netherlands, **3** Department of Family Medicine, Faculty of Medicine, Universiti Kebangsaan Malaysia Medical Centre, Selangor, Malaysia, **4** Neurology Unit, Department of Medicine, Faculty of Medicine, Universiti Putra Malaysia, Selangor, Malaysia

¤ Current address: Department of Social and Preventive Medicine, Faculty of Medicine, University of Malaya, Kuala Lumpur, Malaysia
* angsh.crc@gmail.com

**Data Availability Statement:** All relevant data are presented within the manuscript. Study data are available from Health Informatics Centre, Ministry

## Abstract

### Background and objectives

Risk of readmissions is an important quality indicator for stroke care. Such information is limited among low- and middle-income countries. We assessed the trends for 28-day readmissions after a stroke in Malaysia from 2008 to 2015 and evaluated the causes and factors associated with readmissions in 2015.

### Methods

Using the national hospital admission records database, we included all stroke patients who were discharged alive between 2008 and 2015 for this secondary data analysis. The risk of readmissions was described in proportion and trends. Reasons were coded according to the International Classification of Diseases, 10th Edition. Multivariable logistic regression was performed to identify factors associated with readmissions.

### Results

Among 151729 patients, 11 to 13% were readmitted within 28 days post-discharge from their stroke events each year. The trend was constant for ischemic stroke but decreasing for hemorrhagic stroke. The leading causes for readmissions were recurrent stroke (32.1%), pneumonia (13.0%) and sepsis (4.8%). The risk of 28-day readmission was higher among those with stroke of hemorrhagic (adjusted odds ratio (AOR): 1.52) and subarachnoid hemorrhage (AOR: 2.56) subtypes, and length of index admission >3 days (AOR: 1.48), but lower among younger age groups of 35–64 (AORs: 0.61–0.75), p values <0.001.

### Conclusion

The risk of 28-day readmission remained constant from 2008 to 2015, where one in eight stroke patients required readmission, mainly attributable to preventable causes. Age,

of Health Malaysia. Individual data cannot be made publicly available due to ethical and legal restrictions by the Personal Data Protection Act in Malaysia. Request for aggregate data may be sent to the Institute for Clinical Research at contact@crc.gov.my.

**Funding:** This study was funded by a grant from the Ministry of Health Malaysia – National Institute of Health grant under the project "Monitoring Stroke Burden in Malaysia: the use of linked national data sources" (NMRR-18-100-39847). WYH is supported by the Honours Track of MSc Epidemiology, University Medical Center Utrecht with a grant from the Netherlands Organization for Scientific Research (Grant number: 022.005.021). IV is funded by the Dutch Heart Foundation for project Facts and Figures. The funder had no role in study design, data collection and analysis, decision to publish, or preparation of the manuscript.

**Competing interests:** The authors have declared that no competing interests exist.

ethnicity, stroke subtypes and duration of the index admission influenced the risk of readmission. Efforts should focus on minimizing potentially preventable admissions, especially among those at higher risk.

## Introduction

Despite a decreasing trend in mortality rate from stroke worldwide [1], better survival often translates to higher morbidity and prevalence of chronic stroke. In 2016, there were more than 80 million stroke survivors globally and an estimated loss of 116.4 million disability-adjusted-life-years [2]. The financial burden from having a stroke is also high. About half of the total cost occurred within 30 days of stroke onset. In addition, readmissions within 30 days for complications related to ischemic stroke alone cost nearly US$104 million between 2009 and 2013 [3].

Readmissions have been widely used to measure healthcare performance in management of various diseases including stroke. Rates of readmissions within a month after discharge from a stroke event varied between countries, ranging from 6.5% in Australia to 25% in the United States (US) [4–13]. Majority of the studies related to hospital readmissions were conducted in high-income countries (HICs). While more were studied on readmission rates, few described the temporal trends for readmissions among stroke patients.

There is a scarcity of such information in low- and middle-income countries (LMICs). Having details on hospital readmissions in this region is essential for understanding the local quality of care, improving policy and practice, benchmarking, and serving as baseline estimates for effectiveness of future interventions. This is especially for stroke where its burden is more evident in LMICs [1] and is expected to further increase, with population aging and the rising prevalence of cardiovascular risk factors.

The objective of this study was, therefore, to assess the trends for 28-day risk of readmissions among stroke patients in Malaysia from 2008 to 2015. Furthermore, we determined reasons and risk factors that were associated with the readmissions.

## Methods

### Data source

Data for this secondary data analysis were sourced from the hospital admission database (Sistem Maklumat Rawatan Pesakit) within the Health Information Management System. This database collects all hospital admission records from Ministry of Health (MOH) hospitals. Since 2008, it has included admission records from private hospitals and recently since 2017, from university and Ministry of Defense's hospitals. Due to the period included for this study, only data from public hospitals under the MOH were included in this study.

### Population and definitions

Between 2008 and 2015, all patients with a discharge diagnosis coded to International Classification of Diseases, 10th edition (ICD-10) codes of stroke (I60–I64) and transient ischemic attack (G45) were identified (n = 220532). We included only patients who were discharged alive. Those who absconded, transferred to another hospital, discharged against medical advice, or had unknown discharge statuses were excluded (n = 50897). Furthermore, we excluded patients whose unique identifier was missing, or less than seven characters (n = 3762) since this was the minimum characters needed to identify an individual. Every patient has a unique identifier which is either the national or police identification number, or

passport number. This unique identifier was used together with sex and date of birth to link all admissions that belonged to the same patient within the same year.

Subsequently, the first hospitalization for stroke in each year was tagged as the index admission for that year. Readmission was defined as subsequent readmission within 28 days of discharge from an index admission, including readmission on the same day of discharge. Fig 1 illustrates the process of patient selection.

## Ethical approval

The study was approved by the Medical Research Ethics Committee, Ministry of Health Malaysia (NMRR-18-100-39847) with waiver of informed consent. Patient privacy and confidentiality was protected by using password protected database and restricted access to patient identifiers. Study data were managed in accordance with the Personal Data Protection Act in Malaysia.

## Statistical analysis

The rate of 28-day readmissions for each year was calculated by dividing the number of patients readmitted within 28 days by the total number of index admissions. Stratified by sex, the readmission rates were then age-standardized using the age structure of patients for the year 2015 to assess trends. To obtain percentage change across the years, difference in readmission rates between 2015 and 2008 was divided by the readmission rate in 2008. Mann Kendall's trend test was used to detect the presence of any monotonic trend over the 8-year duration.

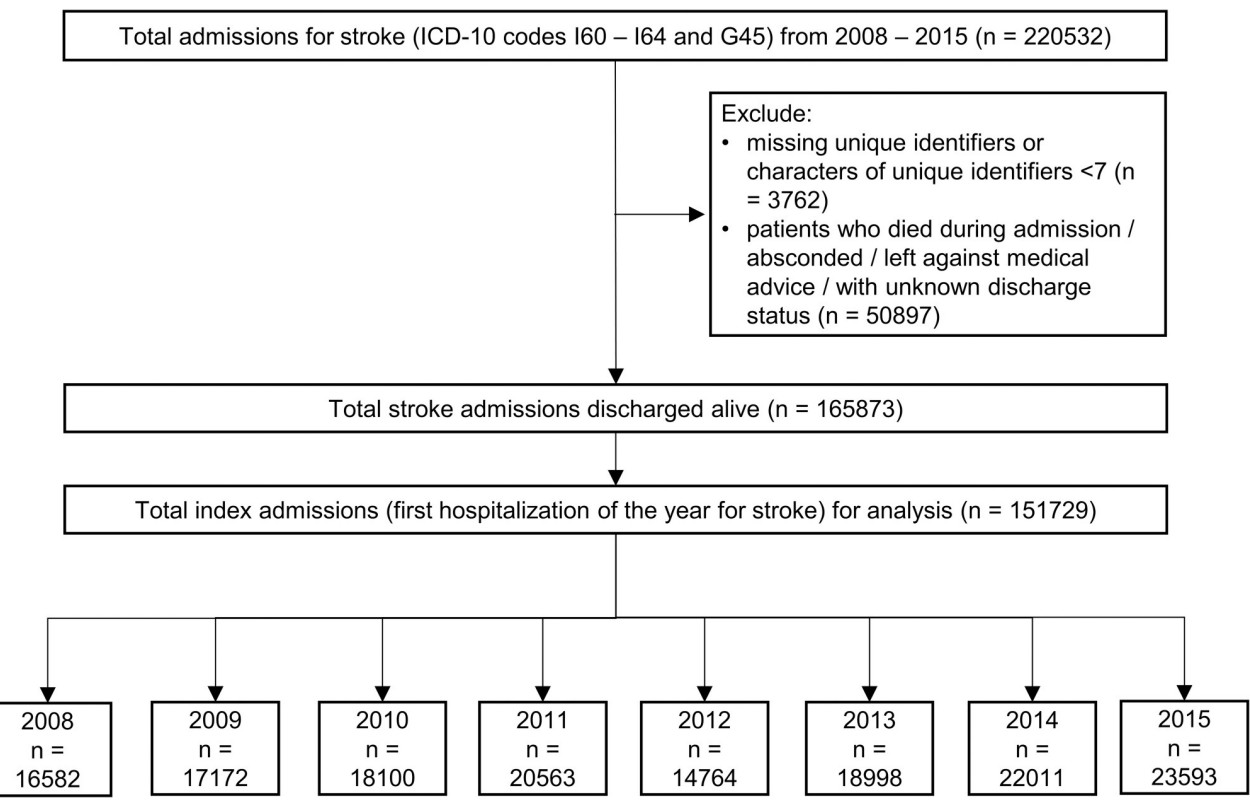

**Fig 1. Flow chart for patient selection.**

Subsequent analyses were restricted to the cohort of stroke patients with readmissions in 2015 because these were the most recent data which could reflect the current clinical care. Causes of readmissions were described in frequencies, where I60–I64 and G45 were grouped as recurrent stroke while other discharge diagnoses were grouped by ICD-10 blocks. To identify factors that were associated with 28-day readmissions, a multivariable logistic regression was conducted. Due to missing values in ethnicity, 86 cases (0.4%) were excluded from the regression. LOS was categorized into ≤3 days and >3 days using the median LOS as the cut-off point. Stroke subtypes were categorized to ischemic stroke (I63, I64, and G45), hemorrhagic stroke (I61 and I62) and subarachnoid hemorrhage (I60). There were no violations of model assumptions or issues with multicollinearity.

All analyses were conducted in R (version 3.5.2) and R Studio (version 1.1.463) [14, 15]. For the regression, "glm2" package was used [16]. Statistical significance was set at $p$ value <0.05.

## Results

### Baseline characteristics

Between 2008 and 2015, a total of 151729 index stroke admissions were included in the analysis. The number of index admissions consistently increased from 16528 in 2008 to 23593 in 2015, except for 2012 and 2013 where a change in the data entry system impeded submission of data from some hospitals. In general, about 60% of the patients were men and nearly two-thirds (61.4–63.5%) were between 50 and 74 years old. Distribution of the ethnicity resembled that of the general Malaysian population, with the majority being Malays (55.3–58.1%). The proportion of stroke cases by subtypes remained similar across the eight years with ischemic stroke contributing to more than 85% of all cases. (Table 1)

### Rates and trends for 28-day readmissions

Between 10.9% and 13.3% of stroke patients were readmitted each year during the study period. By age groups, a rise in readmission rates was seen in men aged above 50 years, except for those between 60 and 69 years old. The largest change was observed among men aged 85 and above (an increase from 10.0% in 2008 to 16.3% in 2015). In contrary, readmission rates decreased among men below 50 years old with the biggest relative reduction being -25.4% at ages 40–44 years. As for women, the greatest increase was noted among the age groups of 55–59 (28.0%) and 85 years and above (27.0%) (Table 2).

Rate of readmissions were slightly higher in women compared to men with absolute differences in the age-standardized rates between men and women ranging from 0.1% to 1.3%. For both sexes, the highest readmission rates were observed among patients with SAH (16.6–22.9% in men and 12.2–31.4% in women), followed by hemorrhagic stroke (15.4–20.6% in men and 13.2–20.9% in women) and ischemic stroke (10.0–11.6% in men and 11.4–13.3% in women) (Table 2).

A plateauing trend was observed for ischemic stroke, with a relative reduction of 1.7% for men and 0.6% for women. There was a notable decreasing trend for hemorrhagic stroke for both sexes over eight years (relative reduction of 10.4% in men and 15.2% in women). On the other hand, SAH showed a 5.4% relative reduction among men in contrast to a 19.4% relative increase in women. (Fig 2, Table 2)

### Causes of readmissions in 2015

The most common diagnosis for a readmission within 28 days after a stroke in 2015 was recurrent stroke (32.1%), followed by complications of stroke such as pneumonia (13.0%), sepsis

**Table 1. Characteristics of stroke patients discharged alive in 2008–2015.**

| | 2008 (n = 16528) | 2009 (n = 17172) | 2010 (n = 18100) | 2011 (n = 20563) | 2012 (n = 14764) | 2013 (n = 18998) | 2014 (n = 22011) | 2015 (n = 23593) |
|---|---|---|---|---|---|---|---|---|
| | n (%) | n (%) | n (%) | n (%) | n (%) | n (%) | n (%) | n (%) |
| Age groups | | | | | | | | |
| 0–34 | 862 (5.2) | 956 (5.6) | 950 (5.2) | 1016 (4.9) | 645 (4.4) | 862 (4.5) | 1085 (4.9) | 1057 (4.5) |
| 35–39 | 374 (2.3) | 419 (2.4) | 437 (2.4) | 483 (2.3) | 354 (2.4) | 456 (2.4) | 636 (2.9) | 694 (2.9) |
| 40–44 | 708 (4.3) | 726 (4.2) | 809 (4.5) | 961 (4.7) | 622 (4.2) | 831 (4.4) | 986 (4.5) | 1152 (4.9) |
| 45–49 | 1219 (7.4) | 1250 (7.3) | 1302 (7.2) | 1481 (7.2) | 1099 (7.4) | 1487 (7.8) | 1667 (7.6) | 1763 (7.5) |
| 50–54 | 1820 (11.0) | 1857 (10.8) | 1951 (10.8) | 2325 (11.3) | 1625 (11.0) | 2064 (10.9) | 2450 (11.1) | 2506 (10.6) |
| 55–59 | 1948 (11.8) | 2099 (12.2) | 2355 (13.0) | 2549 (12.4) | 1870 (12.7) | 2365 (12.4) | 2875 (13.1) | 3045 (12.9) |
| 60–64 | 2152 (13.0) | 2227 (13.0) | 2449 (13.5) | 2765 (13.4) | 1980 (13.4) | 2590 (13.6) | 2959 (13.4) | 3161 (13.4) |
| 65–69 | 2371 (14.3) | 2301 (13.4) | 2344 (13.0) | 2598 (12.6) | 1967 (13.3) | 2455 (12.9) | 2885 (13.1) | 3165 (13.4) |
| 70–74 | 2222 (13.4) | 2337 (13.6) | 2370 (13.1) | 2578 (12.5) | 1816 (12.3) | 2305 (12.1) | 2536 (11.5) | 2621 (11.1) |
| 75–79 | 1479 (8.9) | 1546 (9.0) | 1594 (8.8) | 1935 (9.4) | 1492 (10.1) | 1879 (9.9) | 2140 (9.7) | 2360 (10.0) |
| 80–84 | 878 (5.3) | 894 (5.2) | 995 (5.5) | 1179 (5.7) | 828 (5.6) | 1022 (5.4) | 1128 (5.1) | 1271 (5.4) |
| 85+ | 495 (3.0) | 560 (3.3) | 544 (3.0) | 693 (3.4) | 466 (3.2) | 682 (3.6) | 664 (3.0) | 798 (3.4) |
| Sex | | | | | | | | |
| Men | 9462 (57.2) | 9969 (58.1) | 10339 (57.1) | 11923 (58.0) | 8597 (58.2) | 10946 (57.6) | 12929 (58.7) | 13921 (59.0) |
| Women | 7066 (42.8) | 7203 (41.9) | 7761 (42.9) | 8640 (42.0) | 6167 (41.8) | 8052 (42.4) | 9082 (41.3) | 9672 (41.0) |
| Ethnicity | | | | | | | | |
| Malay | 9144 (55.3) | 9694 (56.5) | 10041 (55.5) | 11602 (56.4) | 8707 (59.0) | 10819 (56.9) | 12757 (58.0) | 13702 (58.1) |
| Chinese | 3670 (22.2) | 3805 (22.2) | 3943 (21.8) | 4480 (21.8) | 2803 (19.0) | 3824 (20.1) | 4137 (18.8) | 4526 (19.2) |
| Indian | 1832 (11.1) | 1812 (10.6) | 1879 (10.4) | 2170 (10.6) | 1283 (8.7) | 1777 (9.4) | 2055 (9.3) | 2101 (8.9) |
| Others [a] | 1882 (11.4) | 1852 (10.8) | 1808 (10.0) | 2286 (11.1) | 1966 (13.3) | 2166 (11.4) | 3037 (13.8) | 3178 (13.5) |
| Unknown | 0 (0.0) | 9 (0.1) | 429 (2.4) | 25 (0.1) | 5 (0.0) | 412 (2.2) | 25 (0.1) | 86 (0.4) |
| Stroke subtypes | | | | | | | | |
| Ischemic [b] | 14570 (88.2) | 15059 (87.7) | 15566 (86.0) | 17545 (85.3) | 12992 (88.0) | 16345 (86.0) | 18836 (85.6) | 20211 (85.7) |
| Hemorrhagic | 1736 (10.5) | 1897 (11.0) | 2261 (12.5) | 2713 (13.2) | 1582 (10.7) | 2298 (12.1) | 2747 (12.5) | 3012 (12.8) |
| SAH | 222 (1.3) | 216 (1.3) | 273 (1.5) | 305 (1.5) | 190 (1.3) | 355 (1.9) | 428 (1.9) | 370 (1.6) |
| Length of stay, median (IQR) | 3 (3.0) | | | | | | | |

SAH, subarachnoid hemorrhage; IQR, interquartile range.

[a] Includes local aborigines and foreigners.

[b] Includes ischaemic stroke (I63), unspecified stroke (I64) and transient ischaemic attack (G45).

(4.8%) and urinary tract infection (UTI) (2.9%). Together with ischemic heart disease (3.7%), these five conditions accounted for more than 50% of readmissions in 2015. (Fig 3) When broken down by subtypes, the main causes were recurrent stroke (29.4%), pneumonia (14.1%) and sepsis (5.1%) among ischemic stroke patients; recurrent stroke (43.6%), pneumonia (9.8%) and injuries to the head (5.0%) among hemorrhagic stroke patients; and recurrent stroke (38.2%), disease of arteries, arterioles and capillaries (6.7%), and other cerebrovascular disease or sequelae (4.5%) among SAH patients. Overall, recurrent stroke remained the most common cause of readmission across all subtypes (Fig 4).

## Factors associated with risk of 28-day readmissions

We found that the odds of being readmitted within 28 days were higher in patients with hemorrhagic stroke (adjusted odds ratio (AOR): 1.52; 95% confidence interval (CI): 1.36–1.69) and

**Table 2. Age-specific and age-standardized readmission rates by sex and stroke subtypes.**

| | 2008 | 2009 | 2010 | 2011 | 2012 | 2013 | 2014 | 2015 | Change [§] | Trend (p-value) [‖] |
|---|---|---|---|---|---|---|---|---|---|---|
| Age groups | | | | | | | | | | |
| **Men** | | | | | | | | | | |
| 0–34 | 14.8 | 17.1 | 16.1 | 14.9 | 14.0 | 11.8 | 15.4 | 14.3 | -3.4 | 0.266 |
| 35–39 | 11.4 | 11.7 | 10.2 | 11.0 | 10.5 | 8.6 | 9.8 | 10.9 | -3.8 | 0.174 |
| 40–44 | 11.8 | 11.6 | 11.8 | 11.1 | 10.0 | 9.3 | 9.4 | 8.8 | -25.4 | 0.004* |
| 45–49 | 10.2 | 11.8 | 10.5 | 11.0 | 8.2 | 8.0 | 10.4 | 8.4 | -17.4 | 0.266 |
| 50–54 | 9.7 | 9.9 | 9.0 | 9.5 | 9.9 | 9.8 | 11.0 | 10.3 | 6.2 | 0.174 |
| 55–59 | 9.8 | 13.3 | 11.2 | 9.5 | 9.9 | 10.4 | 11.2 | 10.1 | 3.3 | 0.902 |
| 60–64 | 12.0 | 11.8 | 12.7 | 11.9 | 10.9 | 10.8 | 11.3 | 10.8 | -9.9 | 0.064 |
| 65–69 | 13.5 | 13.1 | 13.4 | 14.5 | 11.7 | 11.0 | 13.5 | 12.8 | -5.1 | 0.386 |
| 70–74 | 14.0 | 15.0 | 13.4 | 14.1 | 11.9 | 13.1 | 12.8 | 14.6 | 3.9 | 0.536 |
| 75–79 | 14.0 | 13.7 | 13.5 | 16.5 | 11.5 | 13.0 | 14.6 | 14.4 | 2.8 | 0.902 |
| 80–84 | 14.6 | 12.0 | 14.6 | 15.6 | 12.5 | 12.9 | 14.2 | 15.2 | 4.0 | 0.711 |
| 85+ | 10.0 | 12.7 | 16.0 | 15.4 | 14.1 | 14.4 | 16.2 | 16.3 | 63.4 | 0.035 |
| **Women** | | | | | | | | | | |
| 0–34 | 15.0 | 15.7 | 13.9 | 16.8 | 12.3 | 14.5 | 17.3 | 17.2 | 14.5 | 0.386 |
| 35–39 | 9.7 | 10.3 | 12.7 | 12.7 | 8.7 | 14.7 | 11.6 | 8.8 | -8.7 | 0.902 |
| 40–44 | 9.4 | 14.3 | 10.4 | 12.3 | 11.6 | 9.5 | 11.9 | 9.4 | -0.1 | 0.536 |
| 45–49 | 12.9 | 11.8 | 11.5 | 9.6 | 9.7 | 8.8 | 9.7 | 10.3 | -20.4 | 0.108 |
| 50–54 | 10.6 | 9.8 | 12.2 | 8.8 | 12.4 | 8.3 | 11.0 | 10.2 | -4.2 | 0.902 |
| 55–59 | 9.2 | 13.8 | 11.8 | 11.9 | 10.2 | 12.5 | 11.7 | 11.7 | 28.0 | 0.902 |
| 60–64 | 13.0 | 13.8 | 13.7 | 11.5 | 12.4 | 13.5 | 12.9 | 11.5 | -11.3 | 0.266 |
| 65–69 | 15.3 | 12.1 | 14.3 | 14.9 | 11.5 | 14.0 | 15.0 | 14.9 | -2.5 | 0.902 |
| 70–74 | 13.0 | 15.9 | 12.6 | 14.1 | 11.4 | 13.6 | 12.0 | 13.9 | 7.4 | 0.711 |
| 75–79 | 16.0 | 13.5 | 15.3 | 11.3 | 12.0 | 11.8 | 14.2 | 13.4 | -16.2 | 0.386 |
| 80–84 | 13.1 | 13.4 | 14.6 | 15.9 | 13.3 | 10.7 | 12.2 | 14.7 | 12.8 | 0.902 |
| 85+ | 11.5 | 14.4 | 11.7 | 14.1 | 12.1 | 13.8 | 16.8 | 14.6 | 27.0 | 0.108 |
| **Age-standardized readmission rates [†]** | | | | | | | | | | |
| **Men** | | | | | | | | | | |
| Overall | 12.0 | 12.7 | 12.3 | 12.5 | 10.9 | 10.9 | 12.2 | 11.8 | -1.9 | 0.386 |
| Ischemic [‡] | 11.0 | 11.6 | 11.0 | 11.5 | 10.0 | 10.0 | 11.2 | 10.8 | -1.7 | 0.266 |
| Hemorrhagic | 19.2 | 20.6 | 19.3 | 17.4 | 17.1 | 15.4 | 17.2 | 17.2 | -10.4 | 0.064 |
| Subarachnoid hemorrhage | 20.8 | 22.8 | 22.9 | 22.3 | 16.6 | 21.0 | 21.0 | 19.7 | -5.4 | 0.266 |
| **Women** | | | | | | | | | | |
| Overall | 12.8 | 13.3 | 13.1 | 12.6 | 11.6 | 12.2 | 12.9 | 12.7 | -0.2 | 0.386 |
| Ischemic [‡] | 12.1 | 12.6 | 12.1 | 11.8 | 11.4 | 11.6 | 12.1 | 12.0 | -0.6 | 0.266 |
| Hemorrhagic | 18.4 | 18.0 | 20.9 | 17.1 | 13.2 | 14.8 | 17.1 | 15.6 | -15.2 | 0.108 |
| Subarachnoid hemorrhage | 24.4 | 25.3 | 31.4 | 23.7 | 12.2 | 25.1 | 29.7 | 29.1 | 19.4 | 0.711 |

Values presented are percentages (%).

[†] Age was standardized to age structure of patients in 2015.

[‡] Includes ischemic stroke (I63), unspecified stroke (I64) and transient ischemic attack (G45).

[§] Change = (2015–2008) / 2008 x 100%.

[‖] Mann Kendall's trend test for monotonic trend.

* p-value <0.05.

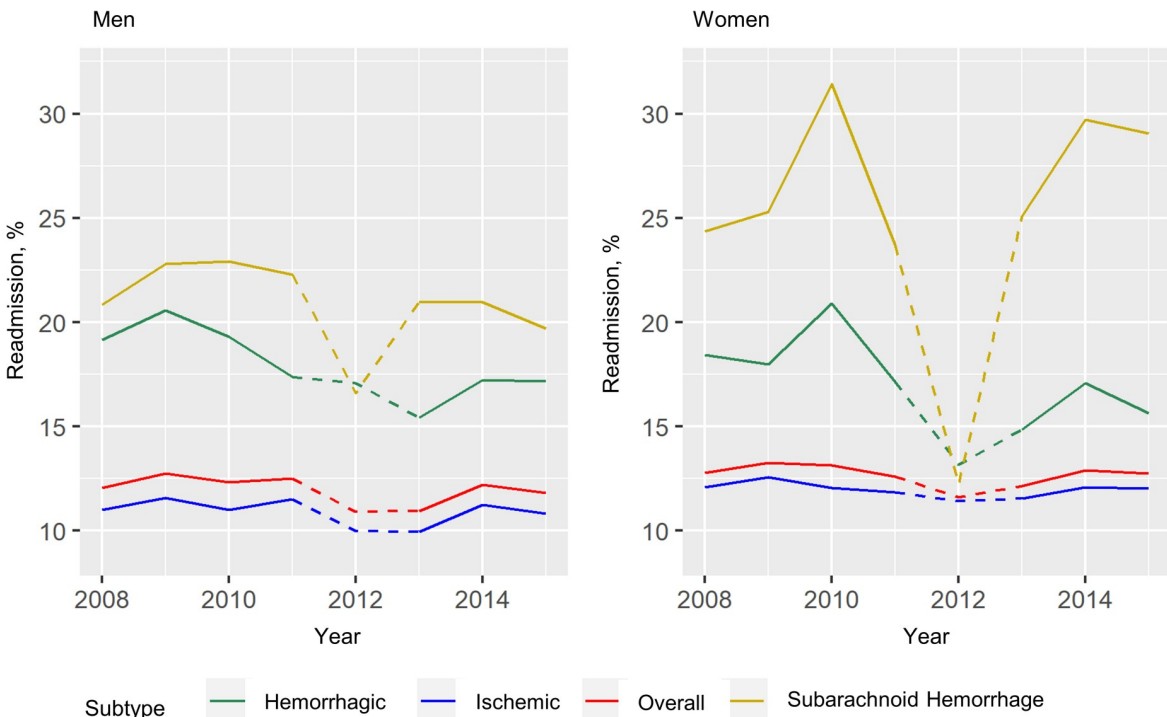

**Fig 2. Age-standardized 28-day readmissions by sex and stroke subtypes.** Dashed lines between 2012 and 2013 indicate loss of some data due to disruption in the flow of data entry during database migration.

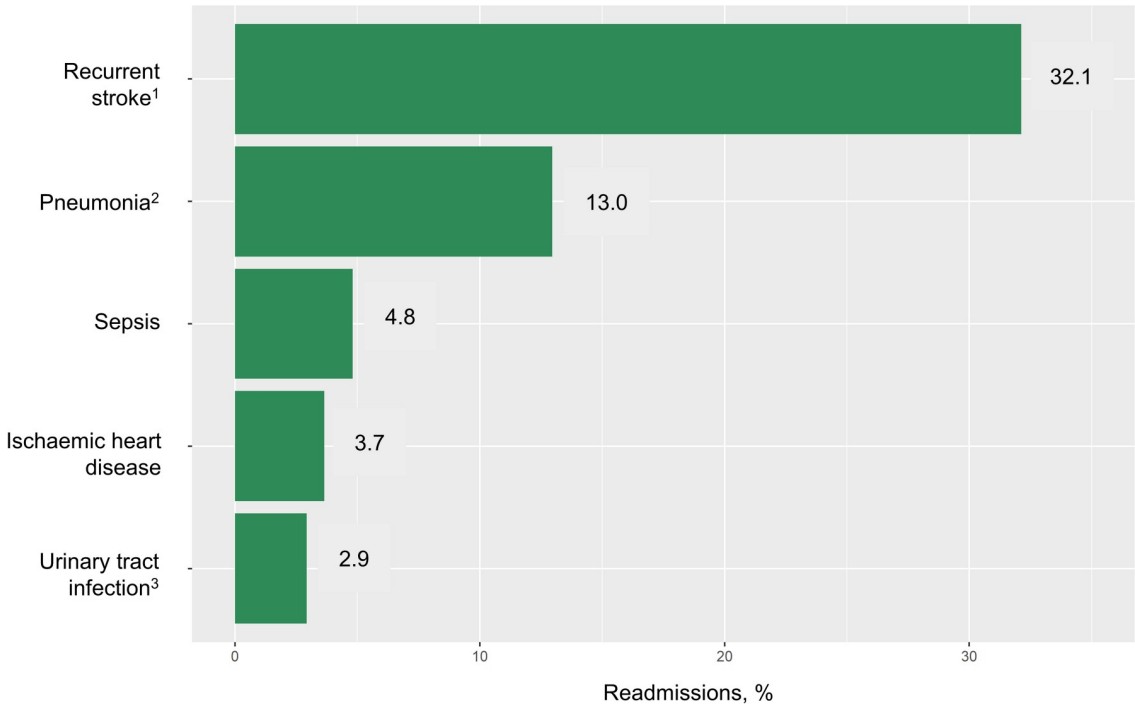

**Fig 3. Top causes of 28-day readmissions in 2015 (n = 2877).** [1] Recurrent stroke includes I60–I64 and G45. [2] Pneumonia also includes J69. [3] Urinary tract infections consisted of N39 (91.5%) and N30–N32 (8.5%).

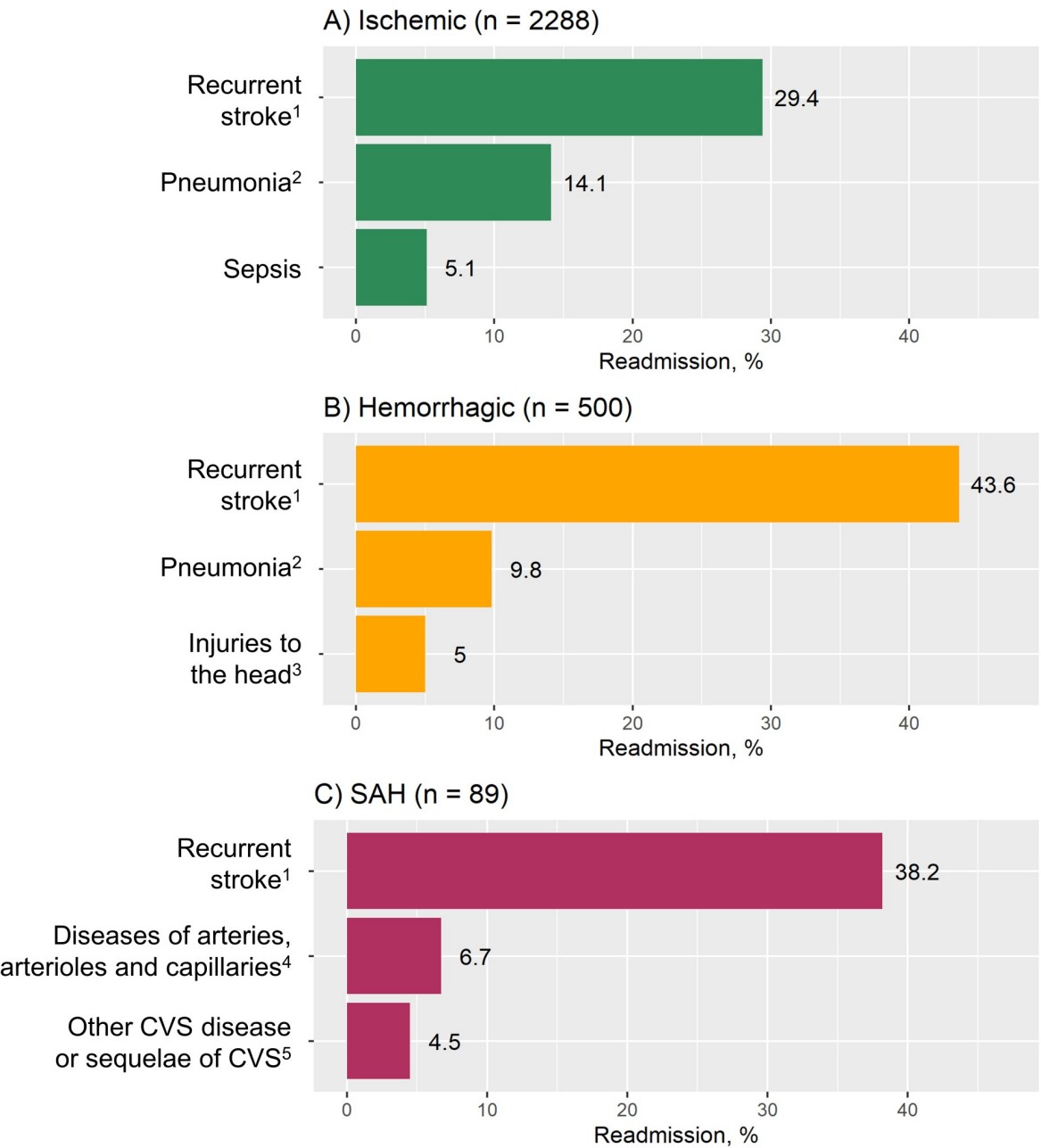

**Fig 4. Top causes of 28-day readmissions in 2015 by stroke subtypes.** A) Ischemic stroke. B) Hemorrhagic stroke. C) Subarachnoid hemorrhage. [1] Recurrent stroke includes I60–I64 and G45. [2] Pneumonia also includes J69. [3] Majority (n = 18, 72.0%) were intracranial injury. [4] Consisted of aortic (n = 1, 16.7%) and other (n = 5, 83.3%) aneurysm and dissection. [5] Consisted of 3 (75.0%) other cerebrovascular disease (CVS) and 1 (25.0%) sequala of CVS.

SAH (AOR: 2.56; 95% CI: 1.99–3.28) when compared to patients who had ischemic stroke during the index admission. Patients with a length of index admission more than 3 days also had 48% higher odds (95% CI: 1.36–1.60) for a readmission than those who stayed 3 days and below. As compared to the oldest age group ($\geq$75), the odds for 28-day readmission declined with decreasing age (Age group: AORs (95% CI) 65–74: 0.99 (0.88–1.11); 55–64: 0.75 (0.66–0.84); 45–54: 0.65 (0.57–0.75); 35–44: 0.61 (0.51–0.73)). On the other hand, patients within the

**Table 3. Multiple logistic regression for factors associated with 28-day readmissions in 2015 (n = 23507 [a]).**

| Factors | Adjusted OR [b] | (95% CI) | *p*-value[*] |
|---|---|---|---|
| **Sex** | | | |
| Men | 1.00 | - | |
| Women | 1.05 | (0.97, 1.14) | 0.224 |
| **Ethnicity** | | | |
| Malay | 1.00 | - | |
| Chinese | 1.02 | (0.92, 1.13) | 0.704 |
| Indian | 1.05 | (0.91, 1.20) | 0.533 |
| Others | 0.86 | (0.76, 0.97) | **0.016** |
| **Age groups** | | | |
| 0–34 | 0.88 | (0.72, 1.07) | 0.202 |
| 35–44 | 0.61 | (0.51, 0.73) | <**0.001** |
| 45–54 | 0.65 | (0.57, 0.75) | <**0.001** |
| 55–64 | 0.75 | (0.66, 0.84) | <**0.001** |
| 65–74 | 0.99 | (0.88, 1.11) | 0.858 |
| **Stroke subtypes** | | | |
| Ischemic stroke [c] (reference) | 1.00 | - | |
| Hemorrhagic stroke | 1.52 | (1.36, 1.69) | <**0.001** |
| Subarachnoid hemorrhage | 2.56 | (1.99, 3.28) | <**0.001** |
| **Length of stay** | | | |
| ≤3 days (reference) | 1.00 | - | |
| >3 days | 1.48 | (1.36, 1.60) | <**0.001** |

OR, odds ratio; CI, confidence interval.

[a] 86 (0.4%) patients were excluded due to missing values in ethnicity.

[b] Adjusted for all variables listed in the table.

[c] Includes ischemic stroke (I63), unspecified stroke (I64) and transient ischemic attack (G45).

[*] Significance set at *p* <0.05.

ethnic group of "Others", who consisted of local aborigines and foreigners, were less likely to be readmitted in comparison to those of Malay ethnicity (AOR: 0.86; 95% CI: 0.76–0.97) (Table 3).

## Discussion

In our study, 1 in 8 stroke patients was readmitted within 28 days of discharge from their index admissions. Over the years, a constant trend of 28-day readmissions was observed for ischemic stroke in contrast to a decreasing trend for hemorrhagic stroke for both sexes. Among those who were readmitted, about one-third were due to recurrent strokes whereas another 21% were caused by stroke-related complications including pneumonia, sepsis and UTI. Age, ethnic groups, stroke subtypes and length of index admission were significantly associated with the odds of 28-day readmissions after a stroke event.

Direct comparison of our findings on readmission rates with rates found in other studies is difficult owing to the differences in study populations, settings and time window to define readmissions. Furthermore, existing studies on stroke readmissions were mostly conducted in HICs. Rates of 30-day readmissions in these countries varied from 6.5% in Australia to 11% in Norway and 12% in the US [7, 13, 17]. In Asia, there was limited information on the rate of readmissions after a stroke discharge. An all-cause 30-day readmission rates of 9% and 10%

was reported in Korea and Taiwan respectively [11, 18]. On the other hand, 29% were readmitted within 31 days for stroke-related conditions in China [19].

While the proportion of 28-day readmissions among patients with ischemic stroke remained unchanged, a drop was observed among patients with hemorrhagic stroke. In contrast, in a US-based study, downward trends were reported for 30-day readmissions after a stroke event for both stroke subtypes, as a result of Hospital Readmissions Reduction Program which emphasized on the post-acute transition of care and regular monitoring on healthcare quality [13]. The different trends for ischemic and hemorrhagic stroke in our study could be due to a few reasons. While the plateauing trend for ischemic stroke is postulated to result from a gap in post-acute or transitional care, shortage in skilled rehabilitation personnel and limited rehabilitation services especially in the community setting, the decreasing trend for hemorrhagic stroke could potentially be explained by an improvement in diagnostic modalities and hyperacute treatment for hemorrhagic stroke which led to better outcomes, fewer complications and subsequently, reduced 28-day readmissions.

In this study, recurrent stroke and infections were the main conditions accounting for over half of all readmissions after a stroke event in 2015. This finding agrees with several other studies. In a meta-analysis, infection (19.9%), coronary artery disease (17.8%) and recurrent stroke (16.0%) were the top three causes of 30-day unplanned readmissions [4]. Within the Asian region, recurrent stroke was the commonest cause for 31-day unplanned readmissions for stroke patients in China [19]. In another study involving 2,657 patients in Taiwan, 34.1% of all 30-day readmissions after a stroke from 2006 to 2008 were caused by infections, and another 16.5% due to recurrent stroke [11]. The stroke-induced immunosuppression renders the patients more susceptible to infections after a stroke event [20]. In addition, common sequelae of stroke such as bladder dysfunction and dysphagia (with subsequent aspiration) predispose stroke patients to urinary tract infections and pneumonia. Quality of care after hospital discharge may also play a role. However, it may be arguable how many of these infections are actually preventable. On the other hand, the high frequency of readmissions due to recurrent stroke could reflect natural disease progression or a gap in secondary prevention of stroke. Prescriptions of antihypertensives drugs and oral anticoagulants among patients with ischemic stroke upon hospital discharge in Malaysia have been found to be suboptimal (48% and 33% respectively) [21].

Older stroke patients, those who stayed more than 3 days during their index admission, and those with a diagnosis of hemorrhagic stroke or SAH had higher risks for 28-day readmissions. Understandably, the abovementioned factors are proxies to stroke severity: older patients are more likely to have a more severe stroke, those with hemorrhagic stroke or SAH often have a greater degree of severity in their stroke and those who stayed longer in the hospital are usually patients who have more severe stroke and are unable to function independently to be discharged home. Furthermore, age and length of stay were common factors found to be associated with increased readmission rate [4, 6, 22]. Older age especially, is associated with a higher chance of having multiple chronic conditions [23], which in turn was shown to increase the risk of readmissions [24].

In addition, we found that certain ethnic groups were at lower odds of 28-day readmissions after a stroke. This included aborigines of the country and foreigners who constituted the majority of an ethnic group "Others". We postulate that access to health care could be more limited for these groups of patients who were often at a disadvantage in terms of socio-economic disparity and thus, leading to fewer hospital readmissions [25, 26]. In agreement to our hypothesis, one study in the US reported that among the minorities, lower admission rates were a result of lack of insurance coverage and poorer access to health care [27]. Nevertheless, contrasting findings were found where Chinese patients in Hawaii were more likely to be readmitted to the hospital compared to whites, attributable to their language barrier [28].

Overall, our results indicated that there is still much room for improvement in our local delivery of stroke care. Firstly, there is a need to reduce acute post stroke complications as the presence of medical complications during initial hospitalizations have been shown to increase the risk of readmissions among ischemic stroke patients [29]. Besides raising awareness on possible types of complications and their timing to healthcare providers, having a stroke-specific care bundle which includes swallowing test, dysphagia assessment, early mobilization and temperature monitoring, as well as provision of care via a stroke unit are other ways to prevent early complications [30]. Parallel to our findings, this should be targeted, in particular among patients who were more likely to suffer from more severe strokes: older patients, those who tend to have longer hospital stay and those who suffer from a hemorrhagic stroke or SAH. Secondly, secondary stroke prevention should be strengthened. Besides local recommendations on the use of pharmacotherapy for secondary stroke prevention [31], the World Health Organization has estimated a 75% reduction of recurrent vascular events with a successful implementation of secondary preventive interventions which involve treatment with secondary preventive drugs and cessation of smoking. In-hospital initiation of secondary stroke prevention has been associated with both short and long term improved vascular outcomes [32]. Thus, strategies to facilitate the implementation of these recommendations and to monitor healthcare providers' adherence to guidelines need to be re-emphasized. Education on the importance of secondary prevention can be enhanced through community engagement [33] to stroke survivors and their care givers. Thirdly, the transition of care for stroke patients from tertiary to primary care has been suboptimal. One of the main issues encountered by primary care clinics that were related to the transfer of care include lack of written information on long term management for stroke post-discharge [34]. Continuity and coordination of post-stroke care can be improved through an integrated care pathway which may include a standardized two-way feedback system between primary, secondary as well as tertiary care levels [35].

To our best knowledge, this study is one of the first to report on trends for readmissions after a stroke in LMICs. We included all eligible cases and there was thus no selection bias. Also, we were able to identify not only readmissions to the same hospital, but other hospitals within MOH too. Admissions to the private hospitals, however, were not included in the analysis. Although we could not claim national representativeness, the results were representative of the public sector which covers nearly 70% of all hospital admissions for the country [36]. Our study was limited by the lack of objective stroke-specific severity measurements, ischemic subtypes and other important clinical data such as comorbidities, functional status and prescription of preventive medication as the data were unavailable in this administrative database. The recurrent events were calculated from the discharge dates of the index events. Furthermore, we acknowledge the underestimations for 2012 and 2013 whereby database migration from a standalone to a web-based system disrupted the data entry resulting in loss of some data.

In conclusion, the risk of 28-day readmissions remained constant from 2008 to 2015. Every 1 in 8 stroke patients was readmitted, mainly attributable to potentially preventable causes. Age, ethnic groups, stroke subtypes and length of the index admission influenced their risk of readmissions. Programs such as stroke-specific care bundle, stroke unit, integrated stroke care pathway, and community engagement and education could be considered to reduce readmissions especially among those at higher risk.

## Acknowledgments

The authors thank the Director General of Health, Ministry of Health Malaysia for his permission to publish these findings. The authors would also like to thank officers from Health Informatic Centre for their support in the provision of the data.

## Disclosure

The manuscript was presented in form of a poster during the 6[th] Asia Pacific Conference on Public Health held on 22 – 25[th] July 2019 in Penang, Malaysia. The abstract for the poster was included in the conference proceeding which was not peer-reviewed [37].

## Author Contributions

**Conceptualization:** Wen Yea Hwong, Michiel L. Bots, Sheamini Sivasampu, Ilonca Vaartjes.

**Formal analysis:** Swee Hung Ang, Wen Yea Hwong.

**Writing – original draft:** Swee Hung Ang, Wen Yea Hwong.

**Writing – review & editing:** Swee Hung Ang, Wen Yea Hwong, Michiel L. Bots, Sheamini Sivasampu, Aznida Firzah Abdul Aziz, Fan Kee Hoo, Ilonca Vaartjes.

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
