## [Decision Letter · Decision Letter 0]

13 Oct 2020

PONE-D-20-28603

Risk of 28-day readmissions among stroke patients in Malaysia (2008 – 2015): trends, causes and its associated factors

PLOS ONE

Dear Dr. Ang,

Thank you for submitting your manuscript to PLOS ONE. After careful consideration, we feel that it has merit but does not fully meet PLOS ONE’s publication criteria as it currently stands. Therefore, we invite you to submit a revised version of the manuscript that addresses the points raised during the review process.

This is a very interesting manuscript in terms of regional analysis of stroke readmissions, which is necessary to understand the behavior of cerebrovascular diseases in Malaysia; unfortunately, deep analysis should be performed to clarify the point addressed by the reviewer, specially in terms of methodology and discussion. Please, also revise English grammar throughout the manuscript.

We look forward to receiving your revised manuscript.

Kind regards,

Miguel A. Barboza, MD, MSc

Academic Editor

PLOS ONE

Journal Requirements:

2. You indicated that you had ethical approval for your study. In your Methods section, please ensure you have also stated whether you obtained consent from participants included in the study or whether the research ethics committee or IRB specifically waived the need for their consent.

3. We noted in your submission details that a portion of your manuscript may have been presented or published elsewhere.

"The manuscript was presented in form of a poster and published in form of an abstract during the 6th Asia Pacific Conference on Public Health held on 22 – 25th July 2019 in Penang, Malaysia."

Please clarify whether this publication was peer-reviewed and formally published. If this work was previously peer-reviewed and published, in the cover letter please provide the reason that this work does not constitute dual publication and should be included in the current manuscript.

Reviewers' comments:

Reviewer's Responses to Questions

**Comments to the Author**

1. Is the manuscript technically sound, and do the data support the conclusions?

Reviewer #1: Yes

2. Has the statistical analysis been performed appropriately and rigorously? 

Reviewer #1: Yes

3. Have the authors made all data underlying the findings in their manuscript fully available?

Reviewer #1: Yes

4. Is the manuscript presented in an intelligible fashion and written in standard English?

Reviewer #1: Yes

5. Review Comments to the Author

Reviewer #1: This is an observational, multicenter study, from a national database, with aim to know trends of early readmission rates in stroke patients in consecutive years, causes, and factors related to early readmissions.

The authors found a “plateau” trend in ischemic stroke, however, a decrease in ICH and SAH were observed. Stroke recurrence and infection complications related represented almost the half of cases of readmissions. Age, ICH, SAH, and length hospital stay were factors associated with admissions. These findings are interesting and relevant in the setting of developing countries.

Just few comments

It would be interesting to know causes of readmission for every stroke subtype and factors associated; the rates of causes could be different according to subtype, and could have implications in strategies and measures to improve it.

Could the author explore the role of stroke severity? I don’t know if scales as NIHSS were collected or registered. The work could gain relevance with this variable playing. If is not possible to get it, should be declared as a limitation.

As the same way, ischemic stroke subtype (by TOAST or others classifications) could add more relevance to the study, knowing that prior recent studies have shown a decreased in stroke recurrence over the years.

I miss a deeper discussion about the findings comparing with prior studies in other regions, mostly, in early readmissions causes.

Please, the references should be verified.

6. PLOS authors have the option to publish the peer review history of their article (what does this mean?). If published, this will include your full peer review and any attached files.

Reviewer #1: No

---

## [Author Response · Author response to Decision Letter 0]

27 Nov 2020

Thank you to the reviewer for the insightful comments. Please see the responses as follow.

1. It would be interesting to know causes of readmission for every stroke subtype and factors associated; the rates of causes could be different according to subtype, and could have implications in strategies and measures to improve it.

Response:

Thank you for the comment. We have added causes of readmission by stroke subtypes in the Result section:

Result, line 194-200, page 11 

When broken down by subtypes, the main causes were recurrent stroke (29.4%), pneumonia (14.1%) and sepsis (5.1%) among ischemic stroke patients; recurrent stroke (43.6%), pneumonia (9.8%) and injuries to the head (5.0%) among hemorrhagic stroke patients; and recurrent stroke (38.2%), disease of arteries, arterioles and capillaries (6.7%), and other cerebrovascular disease or sequelae (4.5%) among SAH patients. Overall, recurrent stroke remained the most common cause of readmission across all subtypes. (Fig 4)

2. Could the author explore the role of stroke severity? I don’t know if scales as NIHSS were collected or registered. The work could gain relevance with this variable playing. If is not possible to get it, should be declared as a limitation. As the same way, ischemic stroke subtype (by TOAST or others classifications) could add more relevance to the study, knowing that prior recent studies have shown a decreased in stroke recurrence over the years.

Response:

Thank you for the suggestion. We agree that stroke severity as well as ischemic subtypes play an important role in determining readmissions of the stroke patients. However, the data used for this study came from an administrative database which did not collect the above information. We have included this as a limitation of our study. It now reads:

Discussion, line 336-338, page 17

Our study was limited by the lack of objective stroke-specific severity measurements, ischemic subtypes and other important clinical data such as comorbidities, functional status and prescription of preventive medication as the data were unavailable in this administrative database.

3. A deeper discussion about the findings comparing with prior studies in other regions, mostly, in early readmissions causes 

Response:

Thank you for the comment. We have added few discussion points on causes of readmissions and rephrase part of the discussion. The references are verified. It now reads:

Discussion, line 272-281, page 14-15

The stroke-induced immunosuppression renders the patients more susceptible to infections after a stroke event. [19] In addition, common sequelae of stroke such as bladder dysfunction and dysphagia (with subsequent aspiration) predispose stroke patients to urinary tract infections and pneumonia. Quality of care after hospital discharge may also play a role. However, it may be arguable how many of these infections are actually preventable. On the other hand, the high frequency of readmissions due to recurrent stroke could reflect natural disease progression or a gap in secondary prevention of stroke. Prescriptions of antihypertensives drugs and oral anticoagulants among patients with ischemic stroke upon hospital discharge in Malaysia have been found to be suboptimal (48% and 33% respectively).[20]

---

## [Decision Letter · Decision Letter 1]

2 Jan 2021

Risk of 28-day readmissions among stroke patients in Malaysia (2008–2015): trends, causes and its associated factors

PONE-D-20-28603R1

Dear Dr. Ang,

We’re pleased to inform you that your manuscript has been judged scientifically suitable for publication and will be formally accepted for publication once it meets all outstanding technical requirements.

Kind regards,

Miguel A. Barboza, MD, MSc

Academic Editor

PLOS ONE

Additional Editor Comments (optional):

Reviewers' comments:

Reviewer's Responses to Questions

**Comments to the Author**

1. If the authors have adequately addressed your comments raised in a previous round of review and you feel that this manuscript is now acceptable for publication, you may indicate that here to bypass the “Comments to the Author” section, enter your conflict of interest statement in the “Confidential to Editor” section, and submit your "Accept" recommendation.

Reviewer #1: All comments have been addressed

2. Is the manuscript technically sound, and do the data support the conclusions?

Reviewer #1: Yes

3. Has the statistical analysis been performed appropriately and rigorously? 

Reviewer #1: Yes

4. Have the authors made all data underlying the findings in their manuscript fully available?

Reviewer #1: Yes

5. Is the manuscript presented in an intelligible fashion and written in standard English?

Reviewer #1: Yes

6. Review Comments to the Author

Reviewer #1: (No Response)

7. PLOS authors have the option to publish the peer review history of their article (what does this mean?). If published, this will include your full peer review and any attached files.

Reviewer #1: No